# Retrieve, Caption, Generate: Visual Grounding for Enhancing Commonsense in Text Generation Models

**Steven Y. Feng**[1]                                                            syfeng@cs.cmu.edu
**Kevin Lu**[2]                                                                  k63lu@uwaterloo.ca
**Zhuofu Tao**[3]                                                               z24tao@g.ucla.edu
**Malihe Alikhani**[4]                                                           malihe@pitt.edu
**Teruko Mitamura**[1]                                                          teruko@cs.cmu.edu
**Eduard Hovy**[1]                                                               hovy@cs.cmu.edu
**Varun Gangal**[1]                                                            vgangal@cs.cmu.edu

[1]*Carnegie Mellon University,* [2]*University of Waterloo*
[3]*University of California Los Angeles,* [4]*University of Pittsburgh*

## Abstract

We investigate the use of multimodal information contained in images as an effective method for enhancing the commonsense of Transformer models for text generation. We perform experiments using BART and T5 on concept-to-text generation, specifically the task of generative commonsense reasoning, or *CommonGen*. We call our approach *VisCTG: Visually Grounded Concept-to-Text Generation*. VisCTG involves captioning images representing appropriate everyday scenarios, and using these captions to enrich and steer the generation process. Comprehensive evaluation and analysis demonstrate that VisCTG noticeably improves model performance while successfully addressing several issues of the baseline generations, including poor commonsense, fluency, and specificity.

## 1. Introduction

Transformer-based models have seen increasing popularity for NLP tasks. This includes SOTA text generation models such as BART [Lewis et al., 2020] and T5 [Raffel et al., 2020]. Model improvements such as larger and better pretrained text generators is a large reason for performance gains. Despite increasing attention on the commonsense reasoning capabilities of models through works like COMET [Bosselut et al., 2019], studies have shown that even large pretrained language models still struggle with commonsense tasks that humans can reason through very easily [Talmor et al., 2019]. We believe that there is commonsense information present in other modalities such as vision, beyond what is present simply in text, which can possibly be used to inject commonsense into text generation models.

In this paper, we show this is true by improving the performance of Transformer-based text generators on concept-to-text generation using visual-grounding, which we call VisCTG: Visually-Grounded Concept-to-Text Generation. Concept-to-text generation is a high-level formulation of several constrained text generation and data-to-text natural language generation (NLG) tasks. These are challenging tasks that have seen increasing interest, and involve generating natural language outputs given certain pre-conditions, e.g. specific words in the outputs, and structured or semi-structured inputs. They typically involve converting a set of inputs into natural language text. These inputs can normally be thought of as

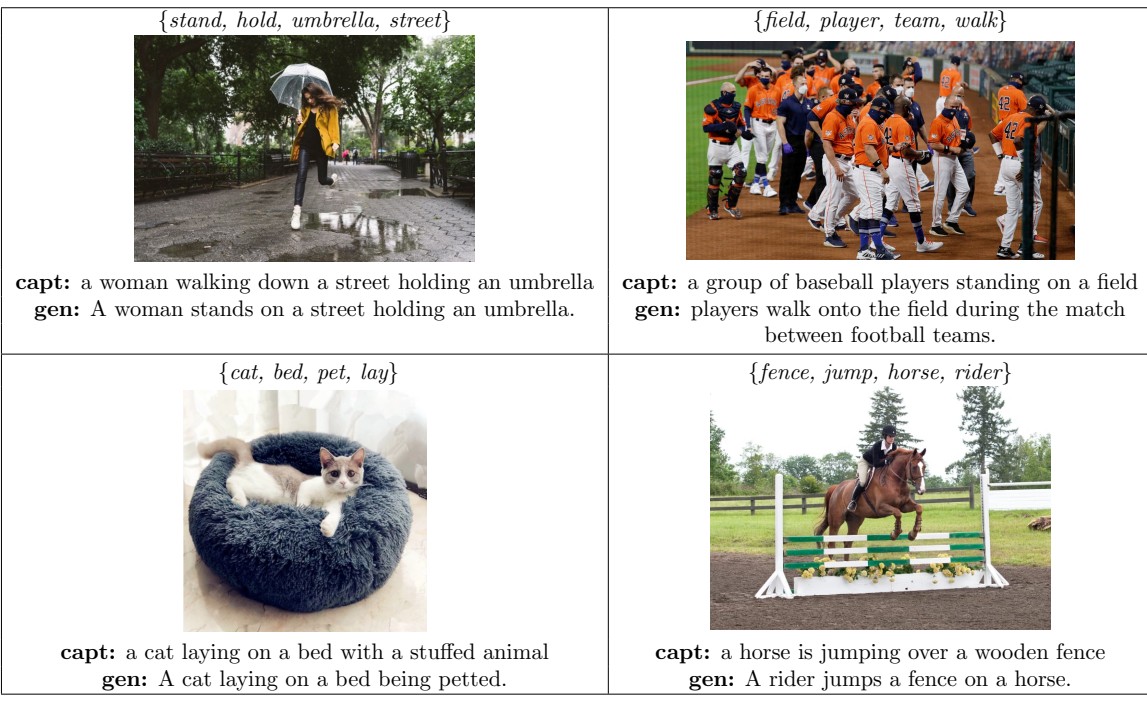

Table 1: Examples of retrieved images for select input concept sets. Corresponding captions and final generations are below the images.

*concepts*, or high-level words or structures, that play an important role in the generated text. Multimodal work has seen increasing popularity, but its exploration for constrained and data-to-text NLG has been limited [Baltrusaitis et al., 2019, Gao et al., 2020].

We investigate the task of generative commonsense reasoning, or CommonGen [Lin et al., 2020], which involves generating sentences that effectively describe everyday scenarios from concepts sets, or words that must appear in the output. CommonGen is challenging as effective relational reasoning ability using commonsense knowledge is required. Models must also possess compositional generalization ability to piece together different concepts. CommonGen is an effective benchmark for constrained text generation and commonsense reasoning as its task formulation and evaluation methodology are rather broadly applicable.

We experiment on CommonGen using different sizes of BART and T5. An initial qualitative analysis (§3.1) of baseline generations shows several issues related to commonsense, specificity, and fluency. We hypothesize that these issues can be addressed through using images and their captions (§3.2). Images representing everyday scenarios are quite commonplace, and are typically logical and grounded in commonsense. Further, image captioning models can normally produce decent captions for everyday images, which can be used to guide and enhance the generation process. See Table 1 for examples.

Expounding on this, we use a pretrained image captioning model on MSCOCO captions [Lin et al., 2015] to caption the top retrieved images for each concept set (§4.1,4.2). We use these captions as additional information to augment inputs to our generation models (§4.3). Extensive evaluation (§6) demonstrates that VisCTG improves model performance while addressing baseline inadequacies with regards to commonsense, specificity, and fluency.

## 2. Dataset, Models, and Metrics

### 2.1 CommonGen Dataset

The original CommonGen dataset is made up of 35,141 concept sets (consisting of 3 to 5 keywords each) and 79,051 sentences, split into train, dev, and test splits. Since the original test set is hidden, we partition the original dev set into new dev and test splits for the majority of our experiments. We do, however, ask the CommonGen authors to evaluate our best VisCTG models on the original test set (more in §6). The training set remains the same. We refer to the original dev and test sets as $dev_O$ and $test_O$, and these new splits as $train_{CG}$, $dev_{CG}$, and $test_{CG}$. Table 2 contains information about these splits. Their relative sizes and distribution of concept set sizes within each are kept similar to the originals.

### 2.2 Models: T5 and BART

We use pretrained text generation models T5 and BART, both base and large versions. Both are seq2seq Transformer models. T5 has strong multitask pretraining. BART is pretrained as a denoising autoencoder to reproduce original from noised text. Their HuggingFace implementations are used for our experiments.

| Dataset Stats | $Train_{CG}$ | $Dev_O$ | $Test_O$ | $Dev_{CG}$ | $Test_{CG}$ |
|---|---|---|---|---|---|
| # concept sets | 32,651 | 993 | 1,497 | 240 | 360 |
| size = 3 | 25,020 | 493 | - | 120 | - |
| size = 4 | 4,240 | 250 | 747 | 60 | 180 |
| size = 5 | 3,391 | 250 | 750 | 60 | 180 |
| # sentences | 67,389 | 4,018 | 7,644 | 984 | 1583 |

Table 2: Statistics of CommonGen dataset splits.

We train two seeded versions of each model on $train_{CG}$ and evaluate their performance on $dev_O$. These serve as the baselines for our experiments. Using the numbers in Lin et al. [2020] as comparison, we validate our implementations. We use the hyperparameters from Lin et al. [2020], beam search for decoding, and select the final epoch as the one reaching maximum ROUGE-2 [Lin and Hovy, 2003] on the dev split. From Table 3, we observe that our re-implementations reach or exceed reported results in Lin et al. [2020] on most metrics.

| Model\Metrics | ROUGE-2/L | | BLEU-3/4 | | METEOR | CIDEr | SPICE | Cov |
|---|---|---|---|---|---|---|---|---|
| Reported BART-large | 22.13 | 43.02 | 37.00 | 27.50 | 31.00 | 14.12 | 30.00 | 97.56 |
| Reported T5-base | 15.33 | 36.20 | 28.10 | 18.00 | 24.60 | 9.73 | 23.40 | 83.77 |
| Reported T5-Large | 21.98 | 44.41 | 40.80 | 30.60 | 31.00 | 15.84 | 31.80 | 97.04 |
| Our BART-base | 15.91 | 36.15 | 38.30 | 28.30 | 30.20 | 15.07 | 30.35 | 93.44 |
| Our BART-large | 17.27 | 37.32 | **39.95** | **30.20** | **31.15** | **15.72** | **31.20** | 95.03 |
| Our T5-base | **17.27** | **37.69** | **41.15** | **31.00** | **31.10** | **16.37** | **32.05** | **94.44** |
| Our T5-large | 17.90 | 38.31 | **43.80** | **33.60** | **32.70** | **17.02** | **33.45** | 96.26 |

Table 3: Comparing $dev_O$ performance of our re-implemented models to those in Lin et al. [2020]. Bold represents where we reach or exceed reported numbers. Results are averaged over two seeds for our models. Lin et al. [2020] did not report BART-base or BERTScore. See §2.3 for evaluation metric explanations.

### 2.3 Evaluation Metrics

We use several evaluation metrics, including those in Lin et al. [2020] such as BLEU [Papineni et al., 2002], CIDEr [Vedantam et al., 2015], SPICE [Anderson et al., 2016], and coverage (cov). These (other than cov) assess token-level similarity between human references and generations. Cov measures the average percentage of concepts covered by the generations.

We also use BERTScore [Zhang et al., 2019] and Perplexity (PPL). BERTScore measures BERT [Devlin et al., 2019] embeddings similarity between individual tokens, serving as a more semantic-level similarity measure. We multiply by 100 when reporting BERTScore.

| Concept Set | Generated Output | Human Reference |
|---|---|---|
| {horse, carriage, draw} | horse drawn in a carriage | The carriage is drawn by the horse. |
| {bathtub, bath, dog, give} | A dog giving a bath in a bathtub. | The teenager made a big mess in the bathtub giving her dog a bath. |
| {dog, house, eat} | A dog eats hay in a house | The dog eats food inside the house. |
| {cow, horse, lasso} | A cow is lassoing a horse. | A group of men riding horses lassoing a cow. |

Table 4: Example generations from our baseline models versus human references.

PPL serves as a measure of fluency, with lower values representing higher fluency. We use GPT-2 [Radford et al., 2019] for PPL. For all other metrics, higher means better performance.

## 3. Initial Analysis and Motivation

### 3.1 Baseline Model Generations

We conduct an initial analysis of the baseline model outputs, and observe that several outputs lack fluency. Some are more like phrases than full coherent sentences, e.g. *"body of water on a raft"*. Others are missing important words, e.g. *"A listening music and dancing in a dark room"* is clearly missing a noun before *listening*. A large portion of generated texts are generic and bland, e.g. *"Someone sits and listens to someone talk"*. This may be an instance of the *dull response problem* faced by generation models [Du and Black, 2019, Li et al., 2015], where they prefer safe, short, and frequent responses independent of the input.

Many generations also appear to lack commonsense. For example, *"body of water on a raft"* is illogical as the phrases *"body of water"* and *"a raft"* are pieced together incorrectly. A similar issue occurs with the {*horse, carriage, draw*} example in Table 4. Further, at times the models cannot understand what certain nouns can or cannot do, e.g. *"A dog checking his phone on a pier."* Several other examples of this can be found in Table 4.

### 3.2 Images and Captions

Images that represent everyday scenarios are quite prevalent for almost any reasonable input concept set. Further, the images are typically grounded in commonsense. For example, searching {*cow, horse, lasso*} will result in many images of cowboys riding horses and lassoing cows, rather than the illogical situation of *"A cow is lassoing a horse."* described by the baseline generation in Table 4. Many everyday images are relatively similar to those in image captioning datasets such as MSCOCO, so pretrained image captioning models should work quite effectively. We thus hypothesize that using images and their captions to visually-ground concept-to-text generation can potentially deal with issues mentioned in 3.1. Retrieved images with corresponding captions generated by a pretrained image captioning model (see §4.2) and final generations for select input concept sets can be found in Table 1.

Textual corpora also suffer from *reporting bias* [Gordon and Van Durme, 2013], where everyday, commonsense albeit "uninteresting" actions (walking), objects (bench) and facts (bananas are yellow) are underrepresented compared to real-world frequency, while "newsworthy" actions (murdering), objects (spaceships) and facts (blue GMO bananas) are exaggerated. This also seeps into large pretrained text models [Shwartz and Choi, 2020]. Using visual data and models dampens this bias, likely improving commonsense of generations.

## 4. Methodology

### 4.1 Image Retrieval

We first obtain images for each concept set in the three splits. Image captioning datasets such as MSCOCO and Flickr are typically too small and focused to be effective since we must cover numerous different concept sets. Further, a search engine is more generalizable.

We decide to use Google Images. On a sample of concept sets, the retrieved images using other search engines were inappropriate; they did not incorporate most input keywords nor handle homonyms well. For example, *"sports+fan+watch"* yields images of fans watching a sports game on Google images, but images of hand watches on Bing and DuckDuckGo.

We queried input concept sets by concatenating keywords with plus signs (+), and used *simple-image-scraper*[1] to obtain URLs of the top 30 resulting images. The image was scraped only if the URL ended in *.png*, *.jpeg*, *.jpg*, or *.gif*. Finally, the received content was verified to be valid images using *pillow*[1], and skipped otherwise. Retrieved images were typically of high quality and corresponded well to the input concepts. See Table 1 for examples.

### 4.2 Image Captioning

After retrieving images, we use a PyTorch-based implementation[2] of the FC image captioning model [Luo et al., 2018, Rennie et al., 2017], which generates a caption sequence via an LSTM model. The LSTM is initialized with a pseudo token, obtained by feeding the input image into a deep CNN followed by a linear projection. For our experiments, we use a pretrained FC model trained on the MSCOCO dataset with pretrained Resnet-101 image features.[2] As most of our retrieved images represent everyday scenarios and are relatively similar to those in MSCOCO, the pre-trained model performs quite well (see example captions in Table 1).

### 4.3 Caption Selection and Input Augmentation

After we have captions $S_c = \{c_1, c_2, ..., c_n\}$ for each concept set in all three splits, we reorder them by descending coverage to the concept set to obtain $S_{c'} = \{c'_1, c'_2, ..., c'_n\}$. If two captions are tied for coverage, we keep them in their original search result order. This allows us to select the captions that have highest coverage and are most relevant.

Since most retrieved images and corresponding captions cover only a fraction of the entire concept set, and the quality of each varies, we hypothesize that using multiple captions for generation may lead to more

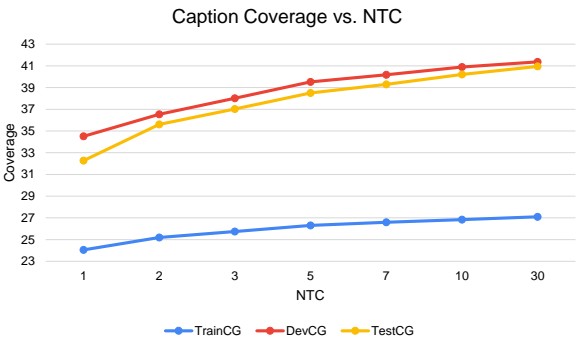

Figure 1: Graph displaying the average coverage by the top NTC captions in aggregate per concept set.

robust and higher-quality outputs with more coverage. The models may learn to piece together information from caption(s) while generating final texts. Hence, we try experiments using different numbers of top captions within $S_{c'}$, a parameter we call *NTC* (Number of

---

1. https://pypi.org/project/simple-image-download/, https://pypi.org/project/Pillow/
2. https://github.com/ruotianluo/self-critical.pytorch, see Appendix A for further model details.

| Augmented Input → Final Generation |
|---|
| wave fall board surfer  a surfer riding a wave on a surfboard → A surfer is falling off his board into the waves. |
| dance stage front crowd  a crowd of people watching a man on a stage  a man is holding a microphone in front of a crowd → A man dances in front of a crowd on stage. |
| stand hold umbrella street  a woman walking down a street holding an umbrella  a woman walking down a street holding an umbrella  a girl holding a pink umbrella in a city  a man holding an umbrella in a city  a group of people standing under a umbrella → A group of people standing on a street holding umbrellas. |

Table 5: Examples of augmented inputs and final generations for varying values of NTC. The augmented inputs are composed of original input keywords followed by  separated augmenting captions.

Top Captions). We try $NTC = 1, 2, 3, 5, 7, 10$, and do not go above $NTC = 10$ as Figure 1 shows that coverage gains from $10 \rightarrow 30$ are minor.

The captions are concatenated together and onto the input concept set using  separator tokens. These serve as augmented inputs to the BART and T5 models. The models learn to convert these augmented inputs to human references during training, and are fed the augmented inputs (corresponding to the value of NTC) during validation and testing. Some examples of augmented inputs and final generations can be found in Table 5.

## 5. Experiments

### 5.1 Model Training and Selection

For training VisCTG models, we mainly follow baseline hyperparameters, barring learning rate (LR) which is tuned per NTC value, and the maximum encoder length which is chosen depending on the tokenizer and value of NTC to ensure the entire input token sequence can fit onto the encoder. We train two seeds per model. See Appendix B for more details.

We choose the epoch corresponding to highest ROUGE-2 on $\text{dev}_{CG}$, and use beam search for decoding. NTC itself is a hyperparameter, so while we train separate versions of each model corresponding to different NTC values, the final chosen models correspond to the NTC values that performed best on $\text{dev}_{CG}$ when averaged over both seeds. We then use the final chosen models to generate on both $\text{test}_{CG}$ and $\text{test}_O$, and report the results in §6.

### 5.2 Human Evaluation

We ask annotators to evaluate 72 $\text{test}_{CG}$ examples, containing the VisCTG and baseline outputs using BART-large and T5-base. These two are chosen as they cover both model types and sizes. See Appendix §C for further details.

We ask annotators to evaluate the text's fluency and commonsense on scales of 1-5. Fluency is a measure of how grammatical, natural, and human-like the text is. Commonsense is the plausibility of the events described by the text. Coverage is more objective compared to fluency and commonsense and we do not evaluate it as the automatic metric suffices.

## 6. Results and Analysis

Automatic evaluation results on $\text{test}_{CG}$ are in Tables 6 and 7, and results on $\text{test}_O$ are in Table 8.[3] Graphs displaying BLEU-4, CIDEr, and SPICE (the metrics on the CommonGen

---

3. We generated on $\text{test}_O$ and got the CommonGen authors to evaluate our outputs on their hidden test set.

| | **BART-base** ($NTC = 5$) | | | **BART-large** ($NTC = 2$) | | |
|---|---|---|---|---|---|---|
| **Metrics** | Baseline | VisCTG | p-value | Baseline | VisCTG | p-value |
| ROUGE-1 | 43.96±0.03 | **45.44**±0.08 | 1.58E-05 | 45.67±0.25 | **46.91**±0.31 | 1.58E-05 |
| ROUGE-2 | 17.31±0.02 | **19.15**±0.21 | 1.58E-05 | 18.77±0.04 | **20.36**±0.05 | 1.58E-05 |
| ROUGE-L | 36.65±0.00 | **38.43**±0.07 | 1.58E-05 | 37.83±0.29 | **39.23**±0.01 | 1.58E-05 |
| BLEU-1 | 73.20±0.28 | **75.65**±0.78 | 6.94E-05 | 74.45±0.21 | **78.80**±0.28 | 6.94E-05 |
| BLEU-2 | 54.50±0.14 | **59.05**±0.07 | 6.94E-05 | 56.25±0.78 | **61.60**±0.85 | 6.94E-05 |
| BLEU-3 | 40.40±0.14 | **44.90**±0.42 | 6.94E-05 | 42.15±0.49 | **47.00**±0.71 | 6.94E-05 |
| BLEU-4 | 30.10±0.14 | **34.10**±0.57 | 3.82E-03 | 32.10±0.42 | **36.25**±0.78 | 2.08E-04 |
| METEOR | 30.35±0.35 | **31.95**±0.07 | 6.94E-05 | 31.70±0.14 | **34.00**±0.14 | 6.94E-05 |
| CIDEr | 15.56±0.10 | **16.84**±0.05 | 6.94E-05 | 16.42±0.09 | **18.35**±0.13 | 6.94E-05 |
| SPICE | 30.05±0.07 | **31.80**±0.28 | 6.94E-05 | 31.85±0.21 | **34.60**±0.28 | 6.94E-05 |
| BERTScore | 59.19±0.32 | **61.44**±0.02 | 1.58E-05 | 59.95±0.29 | **62.85**±0.30 | 1.58E-05 |
| Coverage | 90.43±0.17 | **90.66**±1.39 | 0.33* | 94.49±0.53 | **96.49**±0.24 | 1.58E-05 |
| PPL | 80.39±3.65 | **72.45**±0.79 | 1.58E-05 | 80.37±4.51 | **68.46**±5.90 | 1.58E-05 |

Table 6: Automatic evaluation results (with standard deviations) for BART on test$_{CG}$, averaged over two seeds. Bold corresponds to best performance on that metric per model size. p-value column contains the statistical significance p-values (from Pitman's permutation test [Pitman, 1937]) for VisCTG compared to the corresponding baseline per model size. Insignificant p-values (using $\alpha = 0.05$) are marked with *.

| | **T5-base** ($NTC = 2$) | | | **T5-large** ($NTC = 1$) | | |
|---|---|---|---|---|---|---|
| **Metrics** | Baseline | VisCTG | p-values | Baseline | VisCTG | p-values |
| ROUGE-1 | 44.63±0.13 | **46.26**±0.07 | 1.58E-05 | 46.32±0.26 | **46.93**±0.22 | 7.26E-04 |
| ROUGE-2 | 18.40±0.14 | **19.78**±0.30 | 1.58E-05 | 19.59±0.12 | **20.01**±0.23 | 0.02 |
| ROUGE-L | 37.60±0.16 | **38.91**±0.27 | 1.58E-05 | 39.20±0.21 | **39.52**±0.43 | 0.06* |
| BLEU-1 | 73.60±0.85 | **76.80**±0.28 | 6.94E-05 | 77.55±0.35 | **78.65**±0.21 | 4.65E-03 |
| BLEU-2 | 57.00±0.71 | **60.30**±0.28 | 6.94E-05 | 60.80±0.28 | **61.55**±0.35 | 0.07* |
| BLEU-3 | 42.75±0.49 | **46.25**±0.64 | 6.94E-05 | 46.50±0.00 | **47.10**±0.57 | 0.11* |
| BLEU-4 | 32.70±0.42 | **36.10**±0.85 | 6.94E-05 | 36.20±0.14 | **36.40**±0.28 | 0.21* |
| METEOR | 31.05±0.49 | **32.70**±0.00 | 6.94E-05 | 33.20±0.00 | **33.65**±0.49 | 0.49* |
| CIDEr | 16.26±0.25 | **17.65**±0.02 | 6.94E-05 | 17.79±0.01 | **17.94**±0.25 | 0.23* |
| SPICE | 31.95±0.07 | **33.40**±0.28 | 6.94E-05 | 33.90±0.42 | **34.55**±0.21 | 0.03 |
| BERTScore | 61.40±0.34 | **62.42**±0.17 | 1.58E-05 | 62.67±0.09 | **62.72**±0.03 | 0.34* |
| Coverage | 90.96±1.77 | **94.48**±1.39 | 1.58E-05 | 94.40±0.02 | **95.95**±0.45 | 1.58E-05 |
| PPL | 83.04±1.62 | **77.50**±3.86 | 3.16E-05 | 81.78±4.63 | **73.41**±4.32 | 1.58E-05 |

Table 7: Automatic evaluation results (with standard deviations) for T5 on test$_{CG}$, averaged over two seeds. Bold corresponds to best performance on that metric per model size. p-values column contains the statistical significance p-values (from Pitman's permutation test [Pitman, 1937]) for VisCTG compared to the corresponding baseline per model size. Insignificant p-values (using $\alpha = 0.05$) are marked with *.

leaderboard[4]) on test$_{CG}$ over different values of NTC are in Figure 2. Human evaluation results on test$_{CG}$ are in Table 9. Optimal NTC values for BART-base, BART-large, T5-base, and T5-large are 5, 2, 2, and 1, respectively. These are the VisCTG results reported in the aforementioned tables. Table 10 contains qualitative examples, with more in Appendix §D.

## 6.1 Analysis of Evaluation Results

We see from Tables 6 and 7 that VisCTG outperforms the baselines on all metrics across the models on test$_{CG}$. Performance gains are strong and statistically significant for BART-base, BART-large, and T5-base. VisCTG appears relatively less effective for T5-large which is the strongest baseline, and hence further improving its performance may be more difficult. Table 9 shows that the BART-large and T5-base VisCTG models noticeably outperform their respective baselines in both fluency and commonsense, as rated by human annotators.

---

4. https://inklab.usc.edu/CommonGen/leaderboard.html

| Models\Metrics | ROUGE-2/L | | BLEU-3/4 | | METEOR | CIDEr | SPICE | Coverage |
|---|---|---|---|---|---|---|---|---|
| T5-base (reported baseline) | 14.63 | 34.56 | 28.76 | 18.54 | 23.94 | 9.40 | 19.87 | 76.67 |
| T5-large (reported baseline) | 21.74 | 42.75 | 43.01 | 31.96 | 31.12 | 15.13 | 28.86 | 95.29 |
| BART-large (reported baseline) | 22.02 | 41.78 | 39.52 | 29.01 | 31.83 | 13.98 | 28.00 | 97.35 |
| EKI-BART [Fan et al., 2020] | - | - | - | 35.945 | - | 16.999 | 29.583 | - |
| KG-BART [Liu et al., 2020] | - | - | - | 33.867 | - | 16.927 | 29.634 | - |
| RE-T5 [Wang et al., 2021] | - | - | - | **40.863** | - | **17.663** | **31.079** | - |
| T5-base VisCTG | 22.83 | 44.98 | 45.749 | **34.722** | 31.809 | **16.173** | **28.808** | 92.92 |
| T5-large VisCTG | 23.83 | 45.76 | 47.376 | 36.409 | 33.012 | 16.815 | 29.629 | 95.54 |
| BART-base VisCTG | 21.73 | 43.43 | 43.235 | **32.291** | 30.86 | **15.187** | **27.403** | 88.98 |
| BART-large VisCTG | 23.68 | 45.07 | 48.031 | **36.939** | 33.215 | **17.199** | **29.973** | 94.86 |

Table 8: Automatic evaluation results of VisCTG models on test$_O$ (hidden test set), evaluated by the CommonGen authors. We compare to their reported baseline numbers in Lin et al. [2020] (they did not evaluate BART-base), and models on their leaderboard with publications at the time of writing that outperform the baselines. Their leaderboard only reports BLEU-4, CIDEr, and SPICE. Bold corresponds to best performance (for those three metrics) per model type and size combo.

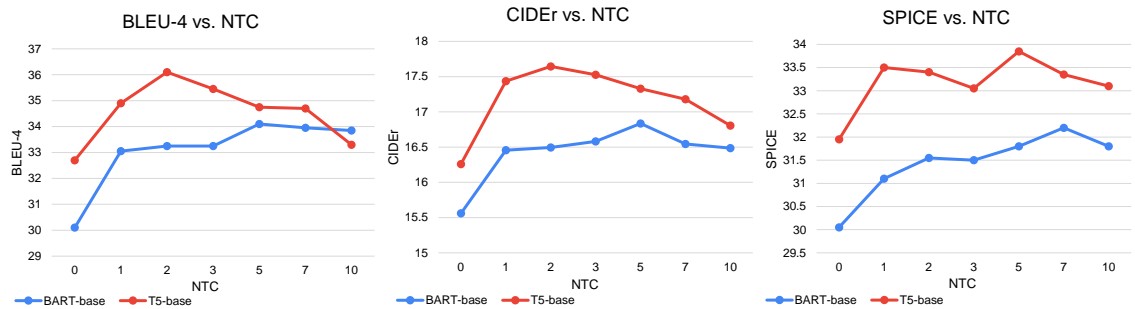

Figure 2: BLEU-4, CIDEr, and SPICE on test$_{CG}$ over different values of NTC for BART-base and T5-base.

From Table 8, we see that VisCTG models substantially outperform corresponding baselines reported in Lin et al. [2020] on test$_O$. T5-base VisCTG outperforms the reported T5-base and large baselines across metrics, and BART-base VisCTG performs similarly to the reported BART-large baseline. BART-large VisCTG

| Model | Method | Fluency | Commonsense |
|---|---|---|---|
| **BART-large** | Baseline | 3.67 | 4.04 |
| | VisCTG | **4.04** | **4.20** |
| **T5-base** | Baseline | 3.82 | 4.18 |
| | VisCTG | **4.25** | **4.65** |

Table 9: Average human eval results on test$_{CG}$, rated on 1-5 scales. Bold corresponds to best performance on that metric for that model. See §5.2 and Appendix C for details.

outperforms the reported baseline, EKI-BART [Fan et al., 2020], and KG-BART [Liu et al., 2020]. These are SOTA published CommonGen BART models that use external knowledge from corpora and KGs. We show that visual-grounding is more effective, and BART-large VisCTG would place very high on the leaderboard.[4] T5-large VisCTG outperforms the reported baseline, but lags behind the SOTA published RE-T5 [Wang et al., 2021].

Figure 2 shows that as NTC increases, BLEU-4, CIDEr, and SPICE increase to a peak, and taper off after. This is expected as we saw in Figure 1 that the rate of increase of coverage declines with larger NTC. The latter images and captions are of diminishing quality, and hence using too many negatively affects model performance.

## 6.2 Qualitative Analysis

Table 10 shows several baseline outputs that contain issues from §3.1, e.g. incomplete/illogical sentences. Human references are all fluent and logical. VisCTG can usually generate much better text than the baselines, addressing issues with fluency, commonsense, and specificity.

The baseline outputs for ex. 1-2 are phrases lacking arguments, and are all illogical for ex. 1-3. Using captions, VisCTG successfully adjusts semantic roles of entities, replaces incorrect subjects, fixes dependency structure, and grounds generations in commonsense. For ex. 1, the captions are of the form "{X} *sitting on a chair with* {Y}", where {X} is a subject and {Y} is an object. VisCTG output has a similar structure, being fluent+logical with higher coverage. For ex. 2, the baseline output treats *"hand of a bird"* as a single entity, the subject. Captions separate *"bird"* and *"hand"* into two, likely guiding the VisCTG output to do so. For ex. 3, the baseline misplaces *"bus"* as subject. Captions are of form "{X} *sitting on a bench* {Y}", where {X} is a logical subject and {Y} is an expression. The VisCTG output has this structure, with correct subject and commonsense, and higher coverage.

For ex. 4, the baseline output lacks a subject that the captions both contain, likely guiding the VisCTG output to contain one: *"a man"*. For ex. 5, the baseline output is generic due to uses of *"someone"*. VisCTG's output is more specific and refers to *"man"*, likely because the caption (although not very fitting) includes a *"man"* subject. Even for captions that fit the concepts less, the structure and fluency can still potentially be exploited.

VisCTG is imperfect. For ex. 6, its output is less logical and lower coverage than the baseline's. The captions are all simplistic and low coverage; the first is illogical, and some others are of the form *"a bunch of apples {...} on a tree"*, likely negatively impacting the generation. Ex. 4's human reference is creative, which is an area where VisCTG still lacks in comparison. For ex. 5, while VisCTG edits *"someone"* to *"man"*, it is unable to merge the two instances of *"man"* or adjust the sentence to be more coherent. These weaknesses are likely because captions tend to be simplistic (due to the captioning model's training data), limiting VisCTG's ability to make heavier edits. VisCTG, unsurprisingly, appears to depend quite heavily on the captions, and hence quality of the images and captioning model.

## 7. Related Work

**Constrained Text Generation:** There have been several works that investigate constrained text generation. Miao et al. [2019] use Metropolis-Hastings sampling to determine Levenshtein edits per generation step, and show gains on several tasks. Feng et al. [2019] propose Semantic Text Exchange to adjust text semantics given a *replacement entity*.

**Data-to-text NLG:** E2E-NLG [Dušek et al., 2018] and WebNLG [Gardent et al., 2017] are two popular NLG benchmarks with structured inputs - meaning representation (MR) and triple sequences, respectively. Montella et al. [2020] use Wiki sentences with parsed OpenIE triples as weak supervision for WebNLG. Tandon et al. [2018] permute input MRs to augment examples for E2E-NLG. Kedzie and McKeown [2019] inject Gaussian noise into a trained decoder's hidden states and sample augmented examples for E2E-NLG.

**Commonsense Reasoning and Incorporation:** Talmor et al. [2019] show that not all pretrained LMs can reason through commonsense tasks. Other works investigate commonsense injection into models; one popular way being knowledge graphs (KGs). One large commonsense KG is COMET, which trains on KG edges to learn connections between words and phrases. COSMIC [Ghosal et al., 2020] is a model that uses COMET to inject commonsense. EKI-BART [Fan et al., 2020] and KG-BART [Liu et al., 2020] use external knowledge (e.g. from corpora and KGs) to improve BART's performance on CommonGen. Distinctly, VisCTG uses visual-grounding and shows higher performance (see §6).

**Multimodal Machine Learning and NLP:** There has been more work on multimodal ML and NLP, in fundamental areas such as multimodal representation and fusion, and application areas such as speech recognition and video captioning, but little for constrained and data-to-text NLG tasks, e.g. CommonGen [Baltrusaitis et al., 2019, Gao et al., 2020].

| Method | Text |
|---|---|
| Concept set | {sit, chair, toy, hand} (example 1) |
| Captions | a little girl sitting on a chair with a teddy bear \<s\> a small child sitting on a chair with a teddy bear \<s\> a young boy sitting on a chair with a skateboard \<s\> a woman sitting on a chair with a teddy bear \<s\> a man sitting on a chair with a remote |
| BART-base-BL | hands sitting on a chair |
| BART-base-VisCTG | A boy sitting on a chair with a toy in his hand. |
| Human | A baby sits on a chair with a toy in one of its hands. |
| Concept set | {food, eat, hand, bird} (example 2) |
| Captions | a bird is perched on a branch with a hand \<s\> a person holding a small bird in their hand |
| BART-large-BL | hand of a bird eating food |
| BART-large-VisCTG | A bird eats food from a hand. |
| Human | A small bird eats food from someone's hand. |
| Concept set | {bench, bus, wait, sit} (example 3) |
| Captions | a man sitting on a bench with a book \<s\> a person sitting on a bench with a laptop |
| T5-base-BL | A bus sits on a bench. |
| T5-base-VisCTG | A man sits on a bench waiting for a bus. |
| Human | The man sat on the bench waiting for the bus. |
| Concept set | {jacket, wear, snow, walk} (example 4) |
| Captions | a young boy in a red jacket is standing in the snow \<s\> a man in a red jacket is standing in the snow |
| BART-large-BL | walking in the snow wearing a furry jacket |
| BART-large-VisCTG | A man is walking in the snow wearing a jacket. |
| Human | Jamie took a walk out into the snow with only a T shirt on and instantly went back inside to wear his jacket. |
| Concept set | {hold, hand, stand, front} (example 5) |
| Captions | a man holding a pair of scissors in front of a wall |
| T5-large-BL | Someone stands in front of someone holding a hand. |
| T5-large-VisCTG | A man stands in front of a man holding a hand. |
| Human | A man stands and holds his hands out in front of him. |
| Concept set | {bag, put, apple, tree, pick} (example 6) |
| Captions | a person holding a apple in a tree \<s\> a bunch of apples are growing on a tree \<s\> a close up of a green apple with a tree \<s\> a bunch of apples are growing on a tree |
| BART-base-BL | A man is putting apples in a bag and picking them up from the tree. |
| BART-base-VisCTG | A man puts a bag of apples on a tree. |
| Human | I picked an apple from the tree and put it in my bag. |

Table 10: Qualitative examples for test$_{CG}$. Color coded final generations: baseline (BL), VisCTG, and human reference. *Concept set* refers to the input keywords and *Captions* refers to the captions (separated by \<s\>) used by the VisCTG model for that particular example to produce its final generation.

## 8. Conclusion and Future Work

In conclusion, we motivated and explored the use of visual grounding for improving the commonsense of Transformer models for text generation. We investigated this for concept-to-text generation, calling our method VisCTG: Visually Grounded Concept-to-Text Generation. Extensive experiments on BART and T5 showed its efficacy on the CommonGen task. Comprehensive evaluation and analysis showed that VisCTG boosts model performance while addressing baseline deficiencies related to commonsense, fluency, and specificity. Potential future work includes improving image search and captioning, e.g. a method for better selection of images during retrieval or using a stronger captioning model. Video captioning and image generation rather than retrieval can also be explored. Further, VisCTG can be investigated for other data-to-text NLG tasks such as WebNLG.

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

## Appendices

### Appendix A. Pretrained FC Image Captioning Model Details

The image encoder is a pretrained Resnet-101 [Ren et al., 2015], where the global average pooling of the final convolutional layer output, a vector of dimension 2048, is taken per image. The spatial features are extracted from the output of a Faster R-CNN [Ren et al., 2015, Anderson et al., 2018] with ResNet-101 [He et al., 2016], trained by object and attribute annotations from Visual Genome [Krishna et al., 2016]. For captioning, the dimensions of LSTM hidden state, image feature embedding, and word embedding are all set to 512. Please see Luo et al. [2018], particularly Sections 3.3 and 5.1, and Rennie et al. [2017], particularly Sections 2 and 5, for more.

### Appendix B. BART and T5 Model Training and Generation Details

T5-large consists of 770M params, T5-base 220M params, BART-large 406M params, and BART-base 139M params. We train two seeded versions of each baseline model and VisCTG model. For all models, we use beam search with a beam size of 5, decoder early stopping, a decoder length penalty of 0.6, maximum encoder lengths of 32 for the baselines and up to 160 for BART and 256 for T5 for VisCTG models, decoder maximum lengths of 32, and a decoder minimum length of 1. For model training, we use a batch size of 64 for T5-base and BART-base, 32 for BART-large, and 8 for T5-large. For T5-base, T5-large, and BART-base, we use 400 warmup steps, and 500 for BART-large. We train all models up to a reasonable number of epochs (e.g. 10 or 20) and perform early stopping using our best judgment (e.g. if metrics have continually decreased for multiple epochs). The learning rates for VisCTG models were determined by trying a range of values (e.g. from 1e-6 to 1e-4), and finding ones which led to good convergence behavior (e.g. validation metrics increase at a decently steady rate and reach max. after a reasonable number of epochs).

Training was done using Google Colab instances which alternately used a single V100 or P100 GPU. The vast majority of the training was done on a single V100 per model. T5-base models trained in approx. 1.5 hours, BART-base models in approx. 1 hour, T5-large models in approx. 6 hours, and BART-large models in approx. 2 hours.

### Appendix C. Human Evaluation Details

The human evaluation study was performed through paid annotators on AMT. Annotators were from Anglophone countries with > 97% approval rate. Each example was evaluated by two annotators. Specific instructions and a question snippet can be seen in Figure 3. Note that T5-base and BART-large studies were conducted separately, so results across both model types are not relatively comparable. The main purpose of the evaluation is to show the relative improvement between the VisCTG models over the corresponding baselines.

The time given for each AMT task instance or HIT was 8 minutes. Sufficient time to read the instructions, as calibrated by authors, was also considered in the maximum time limit for performing each HIT/task. Annotators were paid 98 cents per HIT. The rate of payment (7.35$/hour) exceeded the minimum wage rate for the USA (7.2$/hour) and hence

constitutes fair pay. We neither solicit, record, request or predict any personal information pertaining to the AMT crowdworkers, during the studies.

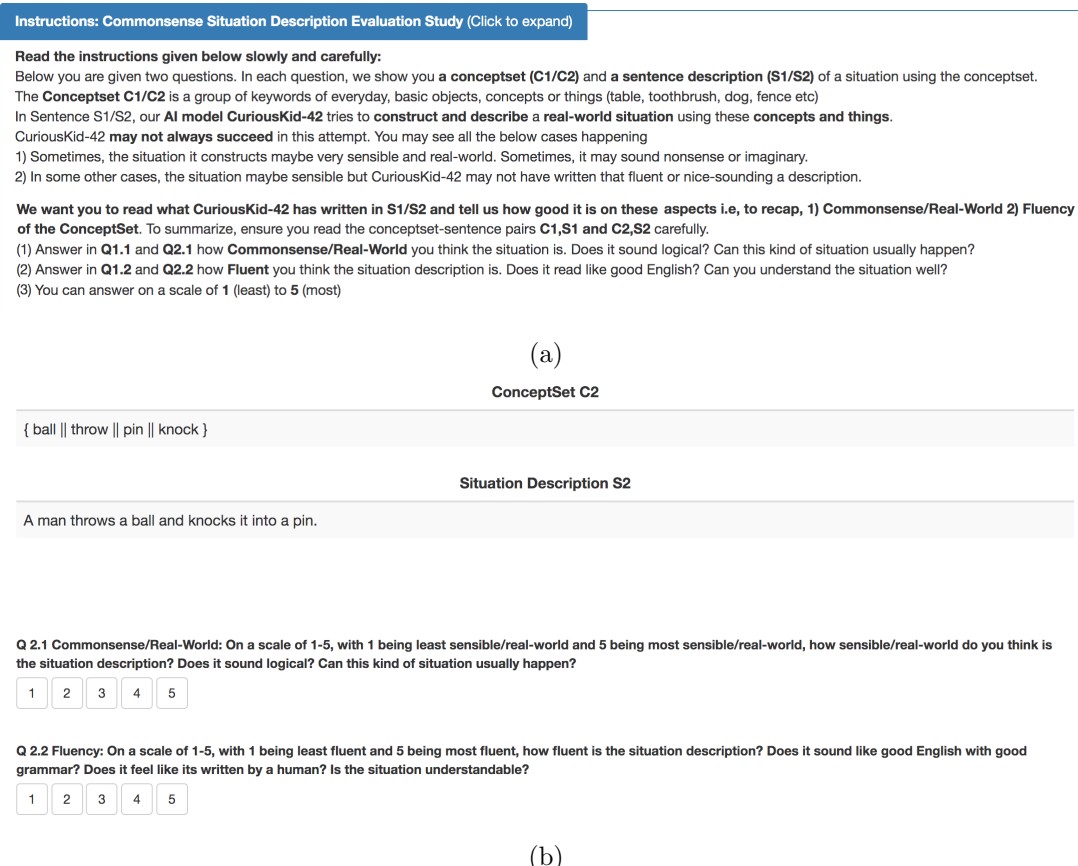

(a)

(b)

Figure 3: Snapshots of human evaluation: a) instructions seen by annotator and b) an example with questions.

## Appendix D. Further Qualitative Examples

See Table 11 for further qualitative examples.

| Method | Text |
|---|---|
| Concept set | {sunglass, wear, lady, sit} |
| Captions | a woman sitting on a bench with a cell phone  a woman sitting on a bench with a book |
| T5-base-BL | A lady sits in a sunglass. |
| T5-base-VisCTG | A lady wearing sunglasses sits on a bench. |
| Human | The lady wants to wear sunglasses, sit, relax, and enjoy her afternoon. |
| Concept set | {music, dance, room, listen} |
| Captions | a person is standing in a room with a bed  a woman is holding a laptop in a room |
| BART-large-BL | A listening music and dancing in a dark room |
| BART-large-VisCTG | A group of people are dancing and listening to music in a room. |
| Human | A boy danced around the room while listening to music. |
| Concept set | {pool, water, slide, slide} |
| Captions | a boat is parked in a water with a boat |
| T5-large-BL | A girl slides into a pool and slides into the water. |
| T5-large-VisCTG | A group of people slide down a slide into a pool of water. |
| Human | A boy slides down a bouncy slide into a pool of water. |
| Concept set | {rock, water, stand, body} |
| Captions | a bird sitting on a rock in a body of water |
| T5-large-BL | a body of water standing on rocks |
| T5-large-VisCTG | A man standing on a rock near a body of water. |
| Human | A bird standing on a large rock in a body of water. |
| Concept set | {card, deck, shuffle, hand} |
| Captions | a person holding a cell phone in their hand  a person holding a pair of scissors in their hand |
| BART-large-BL | a hand shakes a deck of cards |
| BART-large-VisCTG | A man shuffles a deck of cards with his hand. |
| Human | A man shuffles a deck of cards in his hands. |
| Concept set | {chase, ball, owner, dog, throw} |
| Captions | a dog is standing in the grass with a frisbee  a dog is playing with a frisbee in the grass |
| T5-base-BL | owner throws a ball to his dog during a chase. |
| T5-base-VisCTG | A dog is throwing a ball at its owner. |
| Human | The owner threw the ball for the dog to chase after. |
| Concept set | {body, water, bench, sit} |
| Captions | a bench sitting on a beach next to a body of water  a man is sitting on a bench with a cell phone  a bench sitting on a of a beach  a bench sitting in the middle of a lake  woman sitting on a bench with a bird in the background |
| BART-base-BL | A woman sitting on a bench with water in her body. |
| BART-base-VisCTG | A man sits on a bench near a body of water. |
| Human | The woman sat on the bench as she stared at the body of water. |
| Concept set | {bench, sit, talk, phone} |
| Captions | a man sitting on a bench with a cell phone  a woman sitting on a bench with a cell phone  a man sitting on a bench with a cell phone  a person sitting on a bench with a skateboard  a man sitting on a bench with a laptop |
| BART-base-BL | A man sitting on a bench talking to his phone. |
| BART-base-VisCTG | A man sitting on a bench talking on his cell phone. |
| Human | The woman sits on the bench to talk on her daughter on the phone. |

Table 11: Further qualitative examples for test$_{CG}$. Color coded final generations: baseline (BL), VisCTG, and human reference. *Concept set* refers to the input keywords and *Captions* refers to the captions (separated by ) used by the VisCTG model for that particular example to produce its final generation.