# OpenReview forum: "Retrieve, Caption, Generate: Visual Grounding for Enhancing Commonsense in Text Generation Models"
_AKBC.ws/2021/Workshop/CSKB — CSKB_

### Official Review · Reviewer_LjEa · 2021-09-14
**interesting experiments re: adding captions to CommonGen inputs, but did have some concerns about the pitch.**

**Rating:** 6
**Confidence:** 3

**Review:**

The authors augment CommonGen inputs with automatically generated
image captions from images related to the input concept set. Using
T5/BART-base/large, the authors demonstrate that, compared to models
that don't have access to these extra captions, models
perform better according both to human evaluation and to automatic
generation metrics.

The paper tackles an interesting question, that is: what extra
information can visually-grounded captions add to CommonGen? The
assumption the authors have is that some information may only be
contained in images vs. the concept sets (which are generally
represented as text). The paper itself is easy to read and the
experiments were clearly laid out. The conclusion the authors find,
that adding captions can help for most models, is well-supported by
the experiment set.

One concern I have is that the concept sets in CommonGen are
constructed based on images from Flickr30K, MSCOCO, etc., thus, there
may be a bias towards concept sets that are likely to co-occur in
these types of images. While the authors experiments are convincing
for this particular corpus, given that their motivation is address the
textual reporting bias problem from Gordon and Van Durme (2013), I'm
not sure this corpus can support that experiment. The qualitative
explorations, while interesting, don't address the multimodal question
of "what does visual information bring" directly. Furthermore, because
the authors only incorporate visual information via a pretrained image
captioning model, I wonder if 1) a better image captioner vs. that
ResNet model would do better; and 2) something is lost by compressing
images into captions.

There are a few assorted claims I had some issues with:

- On page one, the authors claim that their experiments definitively
  show that visual information is causing better commonsense
  generation. But, while the experiments /suggest/ that could be true,
  there are other reasons why the author's models could be performing
  better. I'd recommend toning the language down re: "we show this is
  true."

- On page five, the authors claim that MSCOCO and Flickr images are
  "too small and focused to be effective since we must cover numerous
  different concept sets." But the concept sets are based on images
  from these corpora, so I'm not sure that this is true for these
  experiments.

Overall, this is an interesting set of experiments that suggests that
adding in automatic image caption generations can be helpful for
CommonGen. I appreciated that both auto and human evals were
undertaken. While this result points towards the idea that extra
information may be contained in images versus just concepts (usually
represented as raw text), the qualitative exploration doesn't fully
convince me

---

### Decision · Program_Chairs · 2021-09-18

Accept